# *TP53* and/or *BRCA1* Mutations Based on CtDNA Analysis as Prognostic Biomarkers for Primary Triple-Negative Breast Cancer

**DOI:** 10.3390/cancers16061184

**Published:** 2024-03-18

**Authors:** Akiko Arimura, Kazuko Sakai, Kazuhisa Kaneshiro, Takafumi Morisaki, Saori Hayashi, Kimihisa Mizoguchi, Mai Yamada, Masaya Kai, Mayumi Ono, Kazuto Nishio, Masafumi Nakamura, Makoto Kubo

**Affiliations:** 1Department of Surgery and Oncology, Graduate School of Medical Sciences, Kyushu University, Fukuoka 812-8582, Japan; 2Department of Surgery, Hamanomachi Hospital, Fukuoka 810-8539, Japan; 3Department of Genome Biology, Kindai University Faculty of Medicine, Osakasayama 589-8511, Japan; 4Department of Breast Surgical Oncology, Kyushu University Hospital, Fukuoka 812-8582, Japan; 5Basic Medical Research Unit, St. Mary’s Research Center, Kurume, Fukuoka 830-8543, Japan

**Keywords:** ctDNA, *TP53*, *BRCA1*, triple-negative breast cancer

## Abstract

**Simple Summary:**

For effective cancer treatment, it is important to scrutinize individual driver gene mutations. In this study, we investigated whether analyzing gene mutations in primary breast cancer based on ctDNA could be a biomarker of prognosis using plasma from 95 patients with primary breast cancer. The key finding was that *TP53* and/or *BRCA1* mutation-positive groups had poor recurrence-free survival in the TNBC patients compared to *TP53* and *BRCA1* mutation-negative groups. This result indicates that *TP53* and/or *BRCA1* mutations on ctDNA could be useful prognostic markers in TNBC patients.

**Abstract:**

Precise biomarkers for predicting the therapeutic efficacy of molecularly targeted drugs are limited at the protein level; thus, it has been important to broadly scrutinize individual cancer driver gene mutations for effective cancer treatments. Multiplex cancer genome profiling can comprehensively identify gene mutations that are therapeutic targets using next-generation sequencing (NGS). In addition, circulating tumor DNA (ctDNA) is a DNA fragment released into the blood by tumor cell-derived cell death or apoptosis. Liquid biopsy with ctDNA is a novel clinical test for identifying genetic mutations in an entire population noninvasively, in real-time, and heterogeneously. Although there are several reports on ctDNA, fewer have evaluated ctDNA with NGS before an initial treatment for breast cancer patients. Therefore, we examined whether analyzing tumor-associated gene mutations in primary breast cancer based on ctDNA could serve as a biomarker for prognosis and optimal treatment selection. Ninety-five primary breast cancer patients treated at our department from January 2017 to October 2020 were included. Pretreatment plasma samples were subjected to NGS analysis of ctDNA, and correlations with patients’ clinicopathological characteristics were evaluated. Fifty-nine (62.1%) patients were positive for ctDNA. ctDNA tended to be positive in hormone receptor-negative, and *TP53* (34%), *BRCA1* (20%), and *BRCA2* (17%) gene mutations were more frequent. Regarding recurrence-free survival, the prognosis was poor in the *TP53* and/or *BRCA1* mutation-positive groups, especially in triple-negative breast cancer (TNBC) patients. In conclusion, the results of this study indicate that ctDNA with liquid biopsy could identify the poor prognosis group before treatment among TNBC patients and for those for whom optimal treatment selection is desirable; additionally, optimal treatment could be selected according to the ctDNA analysis results.

## 1. Introduction

Breast cancer is the most common cancer among women, with an annual incidence of more than 90,000 new patients in Japan, according to the National Cancer Center’s Cancer Registration and Statistics [1] and the National Clinical Database (NCD) registration information from the Japan Breast Cancer Society (JBCS) [2]. However, the five-year survival rate exceeds 90%, which is thought to be due to significant advancements in the understanding of medical check-ups for breast cancer before onset, local therapies such as surgery and radiation therapy, and systemic treatment with drug therapy after onset, based on guidelines such as ASCO [3], ESMO [4], NCCN [5], and JBCS [2]. 

Additionally, breast cancer is broadly classified into subtypes according to the presence or absence of hormone receptors (estrogen receptor (ER) and progesterone receptor (PR)) and human epidermal growth factor receptor 2 (HER2). In particular, triple-negative breast cancer (TNBC) is characterized by a lack of expression of ER, PR, and HER2 and represents up to approximately 20% of all breast cancers. TNBC tumors show no response to endocrine or HER2 targeting agents, leaving chemotherapy as the main systemic course of treatment [6]. In general, TNBC is a high-grade and aggressive disease with a high rate of distant metastasis and is correlated with a poorer outcome compared with other breast cancer subtypes [6]. Additionally, the high heterogeneity of the disease and the lack of uniformly actionable molecular features make it difficult to stratify TNBC patients for tailored treatments [7]. To improve the therapeutic efficacy and prognosis for TNBC patients, it is necessary to establish new treatment strategies and specific biomarkers [8,9].

Many reports say that the comprehensive profiling of driver gene mutations with multiplex cancer genome panel testing, which can be analyzed using next-generation sequencing (NGS), is useful for the clinical practice of recurrent and metastatic solid cancer [10,11,12]. In recent years, two genetic mutation tests have been conducted in general clinical practice to determine the optimal cancer treatment and have been established as a guideline for treatment decision-making in the metastatic setting [13]. Using a companion diagnostic kit tailored to the drug under consideration, single testing with primary or metastatic tissue samples examines the presence or absence of one or a few gene mutations on a single occasion. If a gene mutation is present, treatment is selected based on the guidelines for each cancer that corresponds to that gene mutation. In June 2019, solid cancer patients with recurrence and metastasis were included in insurance coverage in Japan. However, performing multiple tests to identify genetic mutations as therapeutic targets has several problems, such as high cost, insufficient sample size, and a long turnaround time.

Although tissue biopsy is the gold standard in oncological diagnosis, there are several disadvantages. For example, invasive procedures can be performed, and important drivers can be missed due to tumor heterogeneity or distant metastatic lesions [14]. In this regard, liquid biopsy is a promising approach for overcoming these shortcomings as a minimally invasive approach. Circulating tumor DNA (ctDNA) is a single- or double-stranded DNA fragment released from neoplastic cells by apoptosis and necrosis. Recently, ctDNA has been increasingly used as a biomarker to aid in better diagnosis, evaluation of the best treatment, and prognosis of tumor diseases [15,16]. ctDNA based on a liquid biopsy could be more precise than known plasma biomarkers with respect to sensitivity and clinical correlations [17].

For several advanced solid tumors, including genitourinary and gastrointestinal cancers, compared to tissue genotyping, ctDNA genotyping has been shown to significantly shorten the screening duration and improve the trial enrollment rate without compromising treatment efficacy [18]. Reportedly, there is evidence that identifying gene mutations through ctDNA assessment and determining treatment plans based on those results are effective for recurrent breast cancer [19]. For advanced/metastatic breast cancer, there is some evidence that identifying mutations in *ESR1* and *PIK3CA* with liquid biopsy can predict the efficacy of drugs [19,20,21,22]. In early breast cancer, there are reports that analyzing ctDNA after neoadjuvant chemotherapy can predict the recurrence of TNBC [23].

Nevertheless, there are few reports on the evaluation of ctDNA using NGS before initial treatment for all subtypes of breast cancer. By evaluating ctDNA before initial treatment, we believe that poor prognosis groups can be identified, leading to support for personalized medicine. We investigated whether ctDNA, including tumor-related gene mutations in primary breast cancer, could serve as biomarkers for prognosis prediction and optimal treatment selection.

## 2. Materials and Methods

### 2.1. Patients and Samples

We conducted a retrospective study enrolling 95 female patients who had been newly diagnosed with breast cancer and had undergone primary surgery or neoadjuvant chemotherapy (NAC) between January 2017 and October 2020 at Kyushu University Hospital, Fukuoka, Japan. The patients received neoadjuvant treatment according to the National Comprehensive Cancer Network’s (NCCN) Guidelines for the Treatment of Breast Cancer [5]. We reviewed the patients’ electronic medical records and pathological information according to the 8th edition of the Union for International Cancer Control staging system [24]. Peripheral blood samples were obtained before the primary surgery. In NAC cases, blood samples were obtained before NAC and surgery.

Tumor subtypes were evaluated by immunohistochemistry (IHC) staining of surgically resected tissues, as previously reported [25]. Briefly, the classification of ER or PR positivity was defined as ≥1% of tumor cells staining positive for ER or PR, which is called the luminal type. Cancer specimens were defined as HER2-positive when HER2 IHC staining was scored as 3+ according to the standard criteria [26,27] or when HER2 gene amplification was detected using fluorescence spectroscopy with in situ hybridization.

This study conformed to the principles of the Declaration of Helsinki and was approved by the institutional review board of Kyushu University Hospital (no. 2020-591). Before surgery, patients provided comprehensive written consent, which indicated that their medical information could be used for research purposes.

### 2.2. Circulating Tumor DNA Extraction

Peripheral blood samples were analyzed from biobanks in our department, as previously reported [28]. Briefly, blood samples (7–14 mL) were collected from the patients in ethylenediaminetetraacetic acid disodium salt (EDTA) tubes. Blood samples were centrifuged at 1600× *g* for 20 min. The supernatant was subsequently centrifuged within 4 h at 12,000× *g* for 10 min to separate the plasma, after which the supernatant was collected and stored at −80 °C until further use. At least 4 mL of plasma was required for NGS analysis, and those with at least 4 mL were selected from the biobank of our department. DNA was extracted from 4 mL of plasma using an AVENIO ctDNA isolation kit (Roche Diagnostics, Basel, Switzerland) according to the manufacturer’s instructions. The quality and quantity of ctDNA were confirmed using a NanoDrop 2000 device (Thermo Fisher Scientific, Inc., Waltham, MA, USA) and a PicoGreen dsDNA assay kit (Thermo Fisher Scientific, Inc.). The extracted ctDNA was stored at −80 °C until analysis.

### 2.3. Circulating Tumor DNA Sequencing

We used a maximum of 50 ng of DNA for the CAPP-Seq ctDNA analyses using the AVENIO ctDNA Targeted Kit (17 genes; Roche Diagnostics) according to the manufacturer’s instructions. The 17 genes were *ALK*, *APC*, *BRAF*, *BRCA1*, *BRCA2*, *DPYD*, *EGFR*, *ERBB2*, *KIT*, *KRAS*, *MET*, *NRAS*, *PDGFRA*, *RET*, *ROS1*, *TP53*, and *UGT1A1*. The purified libraries were pooled and sequenced on an Illumina NextSeq 500 (Illumina, Inc., San Diego, CA, USA) using a 300-cycle high-output kit. The variants were identified with AVENIO ctDNA analysis software (Roche Diagnostics), which includes bioinformatics methods from CAPP-Seq [29] and integrated digital error suppression [30]. Genetic variants previously cataloged by the Exome Aggregation Consortium at a frequency of 1% were excluded, and only nonsynonymous single nucleotide variants (SNVs), insertions–deletions (Indels), copy number variations (CNVs), and gene fusions involving 17 cancer-related genes were extracted. In addition, using residual samples from NGS analysis, *PIK3CA*, an important molecule in breast cancer, was detected by droplet digital PCR (ddPCR) using the QX200 Droplet Digital PCR System (Bio-Rad Laboratories, Hercules, CA, USA). The probe assay kit was obtained from Bio-Rad Laboratories. We considered mutations to be positive when the variant allele frequency of the detected SNVs was greater than 0.1%. 

### 2.4. Statistical Analysis

Statistical analyses were performed using JMP 16 (SAS Institute, Cary, NC, USA) and Graph Pad Prism version 9.0 (Graph Pad, Inc., SanDiego, CA, USA). The differences in clinical parameters between the groups were evaluated by the analysis of variance for continuous variables, the chi-square test, Fisher’s exact test, the Wilcoxon rank-sum test for categorical variables, and the Cochran–Armitage test for trend analysis. The survival endpoint was recurrence-free survival (RFS), which was defined as the time from the date of breast cancer diagnosis to the date of recurrence and included both local relapse and metastatic disease. Survival curves were generated using the Kaplan–Meier method and compared with the log-rank test. Differences were considered significant when *p*-values were <0.05.

## 3. Results

### 3.1. Flow Chart of Patient Enrollment and Clinicopathological Characteristics Associated with ctDNA Expression

Of the 559 patients, 95 were evaluated after excluding patients with no sample or insufficient biological information, stage 0 or IV breast cancer, or synchronous cancer (Figure 1). The clinicopathological characteristics of all the patients with respect to ctDNA expression are summarized in Table 1. Among the 95 patients, ctDNA expression was classified as positive in 59 (62.1%) and negative in 36 (37.9%). Patients who were ctDNA-positive (62 yo, range 33–91) were significantly older than those who were ctDNA-negative (54 yo, range 33–84) (*p* = 0.0138). Additionally, the proportion with ctDNA positivity was positively correlated with the clinical stage (*p* = 0.0345) but not with tumor size or node status. The proportions of HR-negative patients, including TNBC patients, were greater in the ctDNA-positive patients than in the ctDNA-negative patients (*p* = 0.0059 for HR-negative patients and *p* = 0.0254 for TNBC patients). In addition, examination of tumor markers, carcinoembryonic antigen, and cancer antigen 15–3 revealed a trend toward positive results for ctDNA when either of the tumor markers was positive (*p* = 0.0190). Finally, there was no significant difference between the two groups with respect to nuclear grade, Ki-67 labeling index, or relapse rate.

### 3.2. Profiling of Genetic Mutations in Breast Cancer Using CAPP-Seq

To identify the individual gene mutation profile for each patient, we analyzed ctDNA obtained from pretreatment blood samples by CAPP-Seq using a gene-sequencing panel containing 17 target genes. Among the 95 patients, 59 (62.1%) had one or more genetic mutations. The genetic mutations identified in the patients are summarized in Figure 2. An average of 1.7 mutations were detected per patient (range: 1–5). The frequencies of mutations detected in all patients were as follows: *TP53* (34%), *BRCA1* (20%), *BRCA2* (17%), *ERBB2* (15%), *EGFR* (15%), *PIK3CA* (14%), *ALK* (12%), and *MET* (10%). Other mutations were found in *APC* (8%), *BRAF* (7%), *RET* (5%), *KIT* (5%), *ROS1* (5%), *PDGFRA* (2%), and *NRAS* (2%), all at less than 10%. Among the nine patients with *ERBB2* mutations, SNVs were found in three luminal-type patients, three HER2+-type patients, and one TNBC-type patient. Next, we examined the types of mutations and their frequencies in different subtypes: luminal, HER2+, and TNBC types (Figure 3). The most frequently mutated gene in the luminal type was *BRCA2* (6/38, 16%), and the *BRCA2* mutation rate was greater in this subtype than in the HER2+ (1/33, 3%) and TNBC (2/30, 7%) types. Other mutations in the luminal type were *TP53* (5/38, 13%)*, ALK* (5/38, 13%), *PIK3CA* (4/38, 11%), *BRCA1* (3/38, 8%), *ERBB2* (3/38, 8%), *EGFR* (SNV: 2/38, 5% and CNV: 1/38, 3%), *MET* (CNV: 1/38, 3% and INDEL: 1/38, 3%), *KIT* (2/38, 5%), *ROS1* (2/38, 5%), *BRAF* (1/38, 3%), *RET* (1/38, 3%), and *NRAS* (1/38, 3%). In HER2+ patients, *TP53* (6/33, 18%) was the most common mutation, followed by mutations in *BRCA1* (5/33, 15%), *ERBB2* (SNV: 3/33, 9%, and CNV: 2/33, 6%), *APC* (4/33, 12%), *EGFR* (3/33, 9%), *PIK3CA* (3/33, 9%), *RET* (2/33, 6%), *BRCA2* (1/33, 3%), *ALK* (1/33, 3%), *BRAF* (1/33, 3%), *KIT* (1/33, 3%), and *PDGFRA* (1/33, 3%). Among the TNBC types, *TP53* mutations (9/30, 30%) were detected most frequently, followed by *BRCA1* (4/30, 13%), *MET* (SNV: 3/30, 10% and CNV: 1/30, 3%), *EGFR* (SNV: 2/30, 7% and CNV: 1/30, 3%), *BRCA2* (2/30, 7%), *ALK* (2/30, 7%), *BRAF* (2/30, 7%), *ERBB2* (1/30, 3%), *PIK3CA* (1/30, 3%), *APC* (1/30, 3%), and *ROS1* mutations (1/30, 3%).

### 3.3. Patient Survival

The median follow-up time for this cohort was 1022 days (range 189–1694 days). Moreover, Kaplan–Meier analysis revealed no significant difference in recurrence-free survival (RFS) between ctDNA-positive and ctDNA-negative patients (*p* = 0.5789; Figure 4). There was no significant difference in RFS according to subtype; however, ctDNA-positive patients tended to have a poorer prognosis than ctDNA-negative patients among those with TNBC (not significant, *p* = 0.3975). 

Next, we evaluated the prognostic implications of each genetic variant with RFS as the endpoint. According to the Kaplan-Meier analysis, mutations in *BRCA2*, *ERBB2,* or *PIK3CA* were not significant prognostic factors for RFS. Patients who were positive for *EGFR* tended to have a better prognosis (*p* = 0.3064). However, patients who were positive for *TP53* or *BRCA1* mutations had significantly poorer prognoses than patients who were negative for each mutation (*p* = 0.0054 and *p* = 0.0030, respectively) (Figure 5).

In addition, we evaluated the prognostic value of *TP53* or *BRCA1/2* mutations in different subtypes with Kaplan-Meier analysis (Figure 6). We found that *TP53* or *BRCA1* mutations in the TNBC cohort were significantly associated with a poor prognosis in terms of RFS (*p* = 0.0107 and *p* = 0.0150, respectively). However, neither *TP53* nor *BRCA1/2* mutation was a prognostic factor for the luminal or HER2+ subtype.

### 3.4. Univariate and Multivariate Survival Analysis

Univariate analysis of clinicopathologic characteristics revealed that the subtype (TNBC vs. non-TNBC), *TP53* (mutant vs. wild-type), and *BRCA1* (mutant vs. wild-type) were significantly associated with poor RFS (for TNBC: hazard ratio [HR] = 5.5, 95% confidence interval [CI] = 1.6–19.0, *p* = 0.0075; for *TP53*: HR = 4.0, 95% CI = 1.2–13.9, *p* = 0.028; for *BRCA1*: HR = 5.5, 95% CI = 1.6–19.7, *p* = 0.0083; Table 2A). The individual factors ER/PR status and HER2 status were excluded from the multivariate analysis because these variables were considered in the classification of tumors into the three subgroups. Age at diagnosis, lymph node status, nuclear grade, and Ki-67 index were also excluded from the multivariate analysis through the back elimination method. Multivariate analysis also revealed that TNBC was a prognostic factor for RFS (hazard ratio (HR) = 4.1, 95% CI 1.1–15.2; *p* = 0.038; Table 2B).

## 4. Discussion

The DNA analysis methods used for NGS include whole-genome analysis, whole-exome analysis, and targeted sequencing. The AVENIO ctDNA Analysis System (Targeted Kit) used in this study contains 17 major genes specified in the NCCN guidelines and a target sequence that has been used for research as a treatment selection tool for a wide range of cancer types. In the present study, ctDNA analysis of early-stage breast cancer patients before initial treatment revealed that *TP53* mutations were the most frequently detected mutations, followed by *BRCA1* and *BRCA2* mutations. *TP53* or *BRCA1* mutations were significantly associated with a poor prognosis regarding RFS. Univariate analysis of clinical and pathological features revealed that TNBC type was significantly associated with worse RFS. 

In recent years, treatment strategies based on genome profiling have become increasingly important in cancer therapies. A hospital-based prospective study (TOP-GEAR project, second stage) showed that the percentage of patients treated for druggable mutations was 25 out of 248 (13.4%) patients with metastatic solid tumors [31], and another study showed that 55 out of 423 (13%) patients had metastatic breast cancer in the SAFIR01 cohort [32]. Moreover, ctDNA analysis is a minimally invasive, dynamic, and heterogeneous testing method that has recently attracted increased amounts of attention. One study suggested that liquid biopsy has a greater rate of treatment success than tissue biopsy [18]. A phase 2 trial (plasma MATCH trial) of treatment selection with ctDNA mutations in breast cancer showed the efficacy of neratinib for *ERBB2* mutations and capivasertib for *AKT1* mutations in ctDNA, indicating the usefulness of examining genetic mutations in ctDNA. That study used the Gardant360 gene panel, which took days to test and had a reported detection limit of 0.1% [33]. The panel used in this study targets tumor-related genes that were commonly found in a wide range of cancer types and had a similar detection limit at a lower cost compared to the Gardant360 gene panel. 

According to a previous report, preoperative ctDNA positivity in gastric cancer patients was associated with stage, with more patients having positive ctDNA at stage III than at stages I and II (*p* = 0.0044). Patients with higher tumor stages or lymph node metastases were also more likely to have detectable ctDNA (*p* = 0.005 and *p* = 0.029, respectively) [34]. In this study, ctDNA positivity was also consistent with the clinical stage (*p* = 0.0345) (Table 1). In addition, hormone receptor-negative types, including TNBC, were related to ctDNA positivity (*p* = 0.0059 and *p* = 0.0254, respectively). The association between tumor markers and ctDNA was also examined in this study. Few studies have evaluated the associations between tumor markers and ctDNA in detail [35,36], and the exact factors involved remain unclear. However, in this study, ctDNA positivity was significantly associated with tumor marker (CEA or CA15-3) levels (*p* = 0.0190) (Table 1). Garcia-Murillas I et al. reported that the detection of ctDNA during follow-up is associated with a high risk of future relapse in early-stage breast cancer, regardless of subtype [37]. Before any treatment for any subtype, especially for hormone receptor-negative patients, ctDNA analysis is likely to provide a treatment strategy.

We examined the genomic characteristics of patients with early-stage breast cancer. According to past literature, several genes, including *BRCA1*, *BRCA2*, *CDH1*, *PALB2*, *PTEN*, and *TP53*, were associated with an increased risk of breast cancer. *TP53* is considered the most commonly mutated gene in all cancers, including breast cancer [38]. Other genes that were frequently mutated in breast cancer included *KMT2C*, *KMT2D*, and *ARID1A*, which were involved in epigenetic regulation [38]. Xiao et al. reported that eleven genes, including *ERBB2*, *PIK3CA*, *AKT1*, and *ESR1*, were more frequently associated with mutations or gene amplifications in younger breast cancer patients (≤35 years) [39]. In our study, the most frequently altered genes were *TP53* (34%), *BRCA1* (20%), *BRCA2* (17%), *ERBB2* (15%), *EGFR* (15%), and *PIK3CA* (14%). Overall, the detection rate of high-frequency mutated genes in early breast cancer was relatively stable. Therefore, panel testing with NGS should be modified to take effective drugs for appropriate patients into consideration. In 2012, the Cancer Genome Atlas (TCGA), a public database, reported a detailed genetic analysis of breast cancer using a sample of 825 breast cancer patients [40]. This report described driver genes that were strongly involved in cancer development and growth by subtypes, and somatic mutations in three genes, *TP53*, *PIK3CA*, and *GATA3*, occurred at a frequency of more than 10% in all subtypes. In particular, for TNBC, *TP53* mutations were most frequently observed, which supported our study data. 

A meta-analysis of studies in which ctDNA was detected in breast cancer patients in various settings showed that the detection of ctDNA was significantly correlated with shorter DFS (HR = 4.44, 95% CI 2.29–8.61; *p* < 0.001) [41]. Some studies have reported that ctDNA detection at pretreatment diagnosis is associated with recurrence-free survival (HR = 5.8, 95% CI 1.2–27.1, *p* = 0.01) [37]. Although positive ctDNA at diagnosis was not associated with prognosis in any of the subtypes in this study, *TP53* and/or *BRCA1* mutations detected in ctDNA from TNBC patients were associated with poor prognosis. Mutations in *TP53*, a tumor suppressor gene, are closely related to the proliferation, invasion, and angiogenesis of cancer cells. *TP53* mutations have also been implicated in the poor prognosis of breast cancer patients [42,43,44], which was also observed in the present study. PARP (poly ADP-ribose polymerase) inhibitors are currently used to treat inoperable or recurrent breast cancer that is germline *BRCA* pathogenic/likely pathogenic mutation-positive and HER2-negative with a history of prior cancer chemotherapy [45]. There is no clear information regarding breast cancer patients with mutations in *TP53*, but trials in other cancers have shown the efficacy of WEE1 inhibitors against *TP53* mutations [46,47,48,49,50]. As additional research is conducted on breast cancer, further clinical trials are expected to develop new treatment methods. It is well known that there are fewer novel drugs available for TNBC than for luminal-type breast cancer. Therefore, it could be important that the detection of ctDNA (particularly *TP53* and/or *BRCA1*) at the time of diagnosis before any treatment was associated with the risk of recurrence in this study for patients with TNBC (Table 2A,B). However, further investigation is needed.

Due to the retrospective nature of our study, a few limitations were associated with it. First, the sample size was small because we focused on a limited population in a preliminary clinical setting. Second, only retrospectively collected samples were included, which raises questions about the generalizability of the results. Furthermore, this study had a short observation period (median: 1022 days), which leaves the long-term prognosis unclear. In patients with TNBC, which is known to have a high risk of early recurrence [51,52,53], there seems to be a sufficient observation period. However, for luminal cancer types that can be divided into early and late recurrence groups [54,55], the observation period was considered short. Furthermore, the panel used in this study did not include the *RB* gene, which was thought to predict the efficacy of CDK4/6 inhibitors [56], or the *ESR1* or *PIK3CA* genes, which were frequently observed in breast cancer and were considered involved in endocrine therapy resistance [19,20,21,22,57]. Therefore, additional investigation using ddPCR was required for *PIK3CA*. It is believed that there is room for improvement in genes’ selection for multi-gene targeted kits in future studies.

## 5. Conclusions

Our study revealed that the personalized genomic characteristics of breast cancer patients with ctDNA detectable with liquid biopsy may serve as useful predictive and prognostic indicators. *TP53* and/or *BRCA1* mutations based on ctDNA analysis could be prognostic biomarkers for primary triple-negative breast cancer. Precision oncology with ctDNA-based gene profiling might help establish personalized therapeutic strategies.

## Figures and Tables

**Figure 1 cancers-16-01184-f001:**
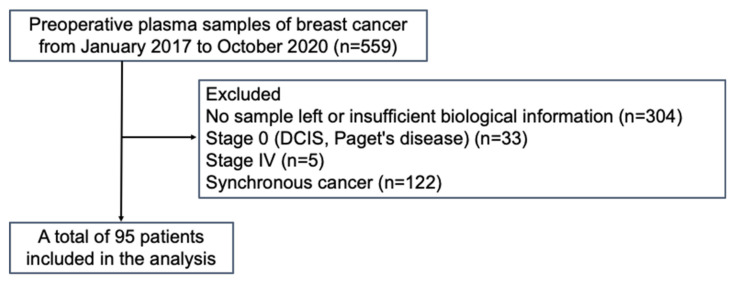
Flowchart of enrolled patients.

**Figure 2 cancers-16-01184-f002:**
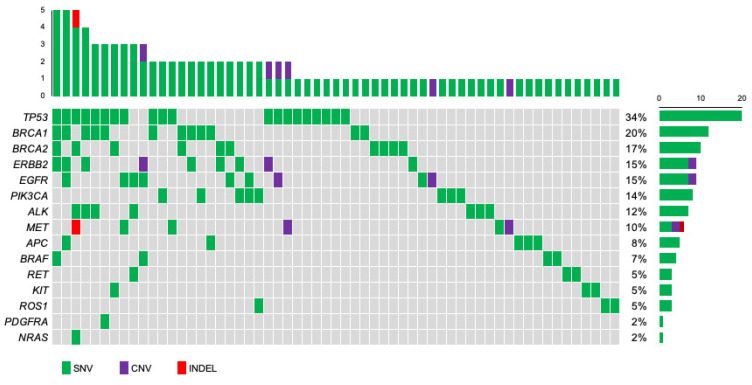
Oncoplot of genomic alterations was identified with next-generation sequencing for 59 primary breast cancer patients. The mutational matrix shows single nucleotide variants (green), copy number variants (purple), and indels (red). **Top**: the number of gene mutations in each patient. **Right**: the percentage of patients with each gene mutation in the total group.

**Figure 3 cancers-16-01184-f003:**
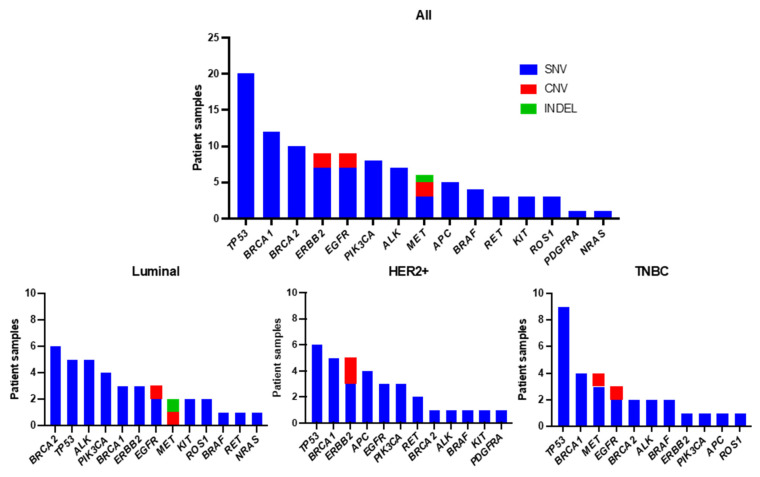
Frequency of detected genetic variants by subtype.

**Figure 4 cancers-16-01184-f004:**
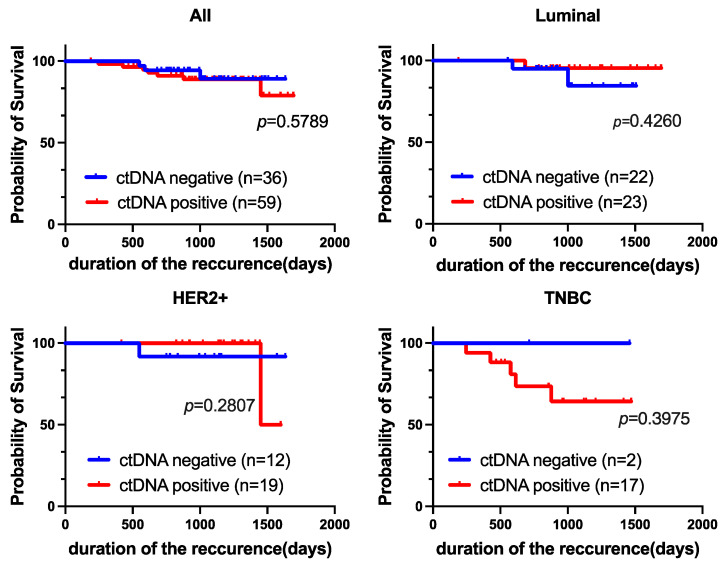
Prognostic value of ctDNA expression. Kaplan-Meier curves showing the estimated RFS based on ctDNA expression. *p*-values are for comparisons of two groups (ctDNA-positive vs. ctDNA-negative).

**Figure 5 cancers-16-01184-f005:**
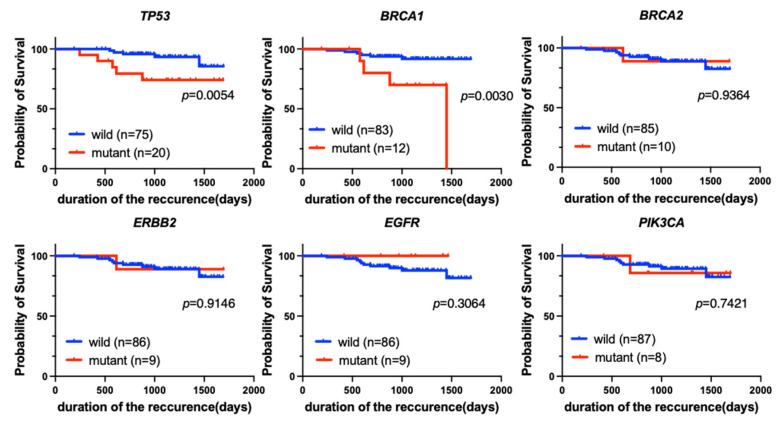
Kaplan-Meier analysis of RFS for patients with each of the top six most common genetic variants (*TP53*, *BRCA1*, *BRCA2*, *ERBB2*, *EGFR,* and *PIK3CA*) among all breast cancer patients (*n* = 95). Patients with *TP53* (*p* = 0.0054) or *BRCA1* (*p* = 0.0030) mutations had significantly worse RFS.

**Figure 6 cancers-16-01184-f006:**
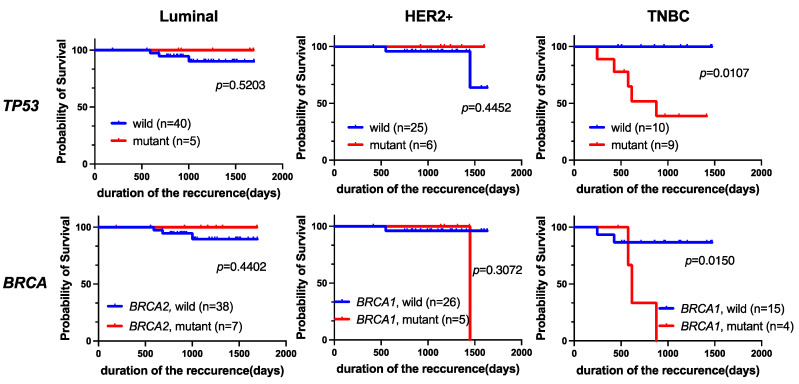
Kaplan-Meier analysis of RFS in breast cancer patients grouped by subtype. Among patients with TNBC, those with *TP53* (*p* = 0.0107) or *BRCA1* (*p* = 0.0150) mutations had significantly worse RFS.

**Table 1 cancers-16-01184-t001:** Clinicopathologic factors are associated with ctDNA positivity.

	ctDNA Positive		ctDNA Negative		*p*
	*n* = 59 (62.1%)		*n* = 36 (37.9%)		
Age at diagnosis					
Median (range)	62	(33–91)	54	(33–84)	0.0138
Tumor size					
T1 (≤2 cm)	21	(35.6%)	14	(38.9%)	0.4880
T2 (>2 cm, ≤5 cm)	30	(50.8%)	18	(50.0%)	
T3 (>5 cm)	5	(8.5%)	4	(11.1%)	
T4	3	(5.1%)			
Nodal status					
Negative	24	(40.7%)	20	(55.6%)	0.1583
Positive	35	(59.3%)	16	(44.4%)	
Clinical stage					
I	16	(27.1%)	12	(33.3%)	0.0345
II	28	(47.5%)	22	(61.1%)	
III	15	(25.4%)	2	(5.6%)	
HR status					
Positive	35	(59.3%)	31	(86.1%)	0.0059
Negative	24	(40.7%)	5	(13.9%)	
HER2 overexpression					
Positive	19	(32.2%)	12	(33.3%)	0.9093
Negative	40	(67.8%)	24	(66.7%)	
Nuclear grade					
1	15	(25.4%)	8	(22.2%)	0.7899
2	13	(22.1%)	9	(25.0%)	
3	30	(50.8%)	19	(52.8%)	
Unknown	1	(1.7%)			
Ki67 levels					0.6962
≤20%	25	(42.4%)	17	(47.2%)	
>20%	33	(55.9%)	19	(52.8%)	
Unknown	1	(1.7%)			
Subtypes					
Luminal	23	(39.0%)	22	(61.1%)	0.0254
Luminal/HER2	12	(20.3%)	9	(25.0%)	
HER2	7	(11.9%)	3	(8.3%)	
TNBC	17	(28.8%)	2	(5.6%)	
Tumor marker					
Positive	23	(39.0%)		(16.7%)	0.0190
Negative	35	(59.3%)		(83.3%)	
Unknown	1	(1.7%)			
Relapse					
Yes	7	(11.9%)	3	(8.3%)	0.5864
No	52	(88.1%)	33	(91.7%)	

**Table 2 cancers-16-01184-t002:** Cox proportional hazards model for recurrence-free survival.

**A. Univariate Analysis**				
		**Recurrence-Free Survival**		
		**HR**	**95% CI**	***p*-Value**
Age	(>50 vs. ≤50)	0.5	0.1–1.7	0.295
Tumor size	(>2 cm vs. ≤2 cm)	5.6	0.7–43.9	0.104
Nodal status	(Positive vs. Negative)	1.4	0.4–5.0	0.599
Nuclear grade	(3 vs. 1 and 2)	1.4	0.4–5.0	0.587
Ki67	(>20% vs. ≤20%)	1.3	0.6–1.5	0.726
HR status	(Positive vs. Negative)	0.3	0.09–1.1	0.072
HER2 status	(Positive vs. Negative)	0.5	0.1–2.2	0.353
Subtype	(TNBC vs. non-TNBC)	5.5	1.6–19.0	0.0075
Tumor marker	(Positive vs. Negative)	1.5	0.4–5.2	0.565
ctDNA	(Positive vs. Negative)	1.4	0.4–5.6	0.581
*TP53*	(mutant vs. wild-type)	4.0	1.2–13.9	0.028
*BRCA1*	(mutant vs. wild-type)	5.5	1.6–19.7	0.0083
**B. Multivariate Analysis**				
		**Recurrence-Free Survival**		
		**HR**	**95% CI**	***p*-Value**
Tumor size	(>2 cm vs. ≤2 cm)	4.7	0.5–42.3	0.165
Nodal status	(Positive vs. Negative)	1.4	0.4–5.3	0.621
Subtype	(TNBC vs. non-TNBC)	4.1	1.1–15.2	0.038
*TP53*	(mutant vs. wild-type)	1.9	0.5–8.0	0.38
*BRCA1*	(mutant vs. wild-type)	2.9	0.7–13.0	0.14

## Data Availability

Data are contained within the article.

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
