# Peer review of "TP53 and/or BRCA1 Mutations Based on CtDNA Analysis as Prognostic Biomarkers for Primary Triple-Negative Breast Cancer"

_cancers, 2024, doi:10.3390/cancers16061184_

Round 1

Reviewer 1 Report

Comments and Suggestions for Authors

The manuscript focused ona retrospective series of TNBC patients tested on a series of liquid biopsy specimens to evaluate prognostic content in TP53 and BRCA1 genes represents a timely relevant and technically correct manuscript available for the publication on this journal after moderate considerations:

- In the introduction section, please, could the authors overview the main challenging points about prognostic startification of TNBC patients?

- In the methodological section, please, could the authros review plasma series selected for molecular testing? How these series may be improved in order to improve clinical significance of molecular data? 

- In the methodological section, the authors also report that a not negligible percentage of cases were negative to ctDNA shedding. Please, could the authors technically discuss how this data may be carried out?

- In the methodological section, the authors also report that PIK3CA molecular testing was performed by adopting a dPCR system? Could the authors show how they harmonized dPCR and NGS derived molecular data? In addition, could the authors discuss about the choice of a NGS panel nto covering PIK3CA gene?

- In the results section, please, could the authors also identify, in accordance with classification criteria, molecular type of BRCA1 alterations found in this cohort? As regards, a difference could be observed on this basis? 

- In addition, lack of clinically relevant of BRCA2 alterations in TNBC patients may suggest a revision of these data because BRCA1/2 are usually considered actors of the same crucial complex for the mainteneance of genomic stability

Comments on the Quality of English Language

Moderate english revision should be approached

Author Response

Thank you very much for taking the time to review and provide constructive comments regarding the improvement of the original manuscript.

Please find the detailed response below and the corresponding revisions in the re-submitted files.

Comments 1: In the introduction section, please, could the authors overview the main challenging points about prognostic stratification of TNBC patients?

Response 1: Thank you for pointing this out. We agree with this comment. Therefore, we have added the points about prognostic stratification of TNBC patients as follows: (Page2 lines 54-65). We have also added four references accordingly.

Besides, breast cancer is broadly classified into subtypes according to the presence or absence of hormone receptors (estrogen receptor (ER) and progesterone receptor (PR)) and human epidermal growth factor receptor 2 (HER2). In particular, triple-negative breast cancer (TNBC) is characterized by a lack of expression of ER, PR and HER2 and represents up to approximately 20% of all breast cancers. TNBC tumors show no response to endocrine or HER2 targeting agents, leaving chemotherapy as the main systemic course of treatment [6]. In general, TNBC is a high-grade and aggressive disease with a high rate of distant metastasis, and is correlated with a poorer outcome compared with other breast cancer subtypes [6]. Additionally, the high heterogeneity of the disease and the lack of uniformly actionable molecular features make it difficult to stratify TNBC patients for tailored treatments [7]. To improve the therapeutic efficacy and prognosis for TNBC patients, it is necessary to establish new treatment strategies and specific biomarkers [8, 9].

Comments 2: In the methodological section, please, could the authors review plasma series selected for molecular testing? How these series may be improved in order to improve clinical significance of molecular data? 

Response 2: Thank you for your comments. We have added the point of plasma series selection as follows: (Page3 lines 133-134). Samples stored less than 4 mL were excluded from this study, because at least 4 mL of plasma was required for NGS analysis. Finally, those that met the requirements were used for analysis, and then the quality and quantity of ctDNA were confirmed. We think that this study was preliminary, so there were some limitations about the number of samples, balance of subtypes, number of investigated genes, type of panel, and so on. Furthermore, we consider that standardizing techniques for blood sampling in the various stages can lead to data standardization. Additionally, integrating data from tissue specimens can provide more comprehensive information, enhancing clinical utility. We aim to continue ongoing evaluation in the near future.

At least 4 mL of plasma was required for NGS analysis, and those with at least 4 mL were selected from the biobank of our department.

Comments 3: In the methodological section, the authors also report that a not negligible percentage of cases were negative to ctDNA shedding. Please, could the authors technically discuss how this data may be carried out?

Response 3: Thank you for your question. I think you are right to wonder why a non-negligible percentage of cases were negative for ctDNA shedding.

In this study, we believe that the ctDNA targeted kit used had a low number of genes on the panel and few genes specific to breast cancer, leading to a high ctDNA negativity rate. Previous reports in the colorectal and lung cancer fields have shown varying ctDNA negativity rates of 28% and 43%, respectively. Additionally, focusing on early-stage (stage I-III) breast cancer patients, previous reports indicated a ctDNA detection rate of 55%. Based on these points, we do not consider the results of this study to indicate low detection rates or poor sensitivity.

  1.  Suzuki, T.; Yoshimura, Y.; Yahata, M.; Yew, P. Y.; Nakamura, T.; Nakamura, Y.; Park, J. H.; Matsuo, R. Detection of circulating tumor DNA in patients of operative colorectal and gastric cancers. Oncotarget 2020, 11 (34), 3198-3207. DOI: 10.18632/oncotarget.27682.
  2. Xie, J.; Yao, W.; Chen, L.; Zhu, W.; Liu, Q.; Geng, G.; Fang, J.; Zhao, Y.; Xiao, L.; Huang, Z.; et al. Plasma ctDNA increases tissue NGS-based detection of therapeutically targetable mutations in lung cancers. BMC Cancer 202323(1), 294. DOI: 10.1186/s12885-023-10674-z.
  3. Bettegowda, C.; Sausen, M.; Leary, R. J.; Kinde, I.; Wang, Y.; Agrawal, N.; Bartlett, B. R.; Wang, H.; Luber, B.; Alani, R. M.; et al. Detection of circulating tumor DNA in early- and late-stage human malignancies. Sci Transl Med 2014, 6 (224), 224ra224. DOI: 10.1126/scitranslmed.3007094.

Comments 4: In the methodological section, the authors also report that PIK3CA molecular testing was performed by adopting a dPCR system? Could the authors show how they harmonized dPCR and NGS derived molecular data? In addition, could the authors discuss about the choice of a NGS panel nto covering PIK3CA gene?

Response 4: Thank you for this suggestion. We have added the method about ddPCR for PIK3CA molecular testing as follows: (Page4 lines 152). In the ready-made, relatively-not-expensive, and minimal panel for all the common tumors that was used in this study, PIK3CA was not included, but there are also panels developed for clinical research purposes, such as the Oncomine Breast ctDNA assay. Therefore, PIK3CA, an important molecule in breast cancer, was confirmed by using ddPCR. Finally, PIK3CA mutation in ctDNA was not a useful prognostic biomarker.

In addition, PIK3CA, which was not included in the panel, but an important molecule in breast cancer, was detected by droplet digital PCR (ddPCR) using the QX200 Droplet Digital PCR System (Bio-Rad Laboratories, Hercules, CA, USA).

Comments 5: In the results section, please, could the authors also identify, in accordance with classification criteria, molecular type of BRCA1 alterations found in this cohort? As regards, a difference could be observed on this basis? 

 In addition, lack of clinically relevant of BRCA2 alterations in TNBC patients may suggest a revision of these data because BRCA1/2 are usually considered actors of the same crucial complex for the mainteneance of genomic stability

Response 5: Thank you for your comments. According to a previous report that assessed clinical subtypes in patients with BRCA1/2 pathogenic variants, of 177 breast cancer patients with the BRCA1 variant, 147 (83.1%) were TNBC and 30 (16.9%) were luminal type. Whereas, of 122 breast cancer patients in the BRCA2 variant, 25(20.5%) were TNBC and 97 (79.5%) were luminal type, respectively [1]. In our study, of 12 breast cancer patients with the BRCA1 mutation, 4 (33%) were TNBC and 3 (25%) were of the luminal type. 5 (42%) of the BRCA1 mutation were HER2-positive, which differed from the previous reports where the coexistence of BRCA mutation and HER2 gene amplification was rare with rates ranging from 0-10% [2]. We consider the possibility that limitations such as the balance of subtypes and the number of specimens may have affected the results. Of 10 breast cancer patients with the BRCA2 mutation, 2 (20%) were TNBC, 7 (70%) were luminal type, and 1 (10%) was HER2 type, which were generally consistent for the previous data. We also examined mutation sites, but there was no novel and specific significance regarding BRCA2-mutated TNBC according to the ClinVer.

  1. Okano, M.; Nomizu, T.; Tachibana, K.; Nagatsuka, M.; Matsuzaki, M.; Katagata, N.; Ohtake, T.; Yokoyama, S.; Arai, M.; Nakamura, S. The relationship between BRCA-associated breast cancer and age factors: an analysis of the Japanese HBOC consortium database. J Hum Genet 2021, 66 (3), 307-314. DOI: 10.1038/s10038-020-00849-y.
  2. Tomasello, G.; Gambini, D.; Petrelli, F.; Azzollini, J.; Arcanà, C.; Ghidini, M.; Peissel, B.; Manoukian, S.; Garrone, O. Characterization of the HER2 status in BRCA-mutated breast cancer: a single institutional series and systematic review with pooled analysis. ESMO Open 2022, 7 (4), 100531. DOI: 10.1016/j.esmoop.2022.100531.

Reviewer 2 Report

Comments and Suggestions for Authors

The authours of this study entitled 'TP53 and/or BRCA1 Mutations based on ctDNA Analysis as Prognostic Biomarkers for Primary Triple-Negative Breast Cancer' describe the analysis of circulating tumoural DNA in breast cancer patients and its associations with patient outcomes. Interestingly, the authours show that a number of genes, namely TP53, BRCA1 and PIK3CA that are frequently mutated in breast tumours are also detected in circulating tumour DNA, suggesting the validity of this approach. Moreover, their presence in the blood associates with poor outcome of the patients, similarly to what has been shown for breast tumours. Considering the fact that collecting circulating tumour DNA is a non-invasive approach, this strategy might help guide the treatment options in the clinic. Overall, this article describes new and interesting piece of informations regarding breast cancer and therefore it is fit for publication in its present form.

Author Response

Thank you very much for providing important comments. We are thankful for the time and energy you expended. 

Reviewer 3 Report

Comments and Suggestions for Authors

The manuscript by Shimazaki et al. describes the employment of a commercial kit to isolate and identify sequences of circulating tumor DNA. In the samples collected and screened for this work, the most present mutations for TNBC were BRAC2 and TP53 and this finding gives name to the work. Thus, this reviewer suggests a few changes in the manuscript, which are listed below.

major:

Please include a comparison with the average findings in Japanese and other populations in the discussion section using cancer databanks information.

minor: 

Please include a sentence in the abstract section defining Circulating tumor DNA (ctDNA).

A few words throughout the text are not separated by space. Please check.

Author Response

Thank you very much for taking the time to review and provide constructive comments regarding the improvement of the original manuscript.

Please find the detailed response below and the corresponding revisions in the re-submitted files.

Comments 1: Please include a comparison with the average findings in Japanese and other populations in the discussion section using cancer databanks information.

Response 1: Thank you for pointing this out. We agree with this comment. Therefore, we have added a comparison with the average findings using cancer databanks information as follows:

(Page13 lines 393-398).

In 2012, the Cancer Genome Atlas (TCGA) revealed a detailed genetic analysis of breast cancer using a sample of 825 breast cancer patients [40]. This report described driver genes that were strongly involved in cancer development and growth by subtypes, and somatic mutations in three geners, TP53, PIK3CA, and GATA3, occurred at a frequency of more than 10% in all subtypes. In particular, for TNBC, TP53 mutations were most frequently observed, which supported our study data.  

We have also added one references accordingly.

Comments 2: Please include a sentence in the abstract section defining Circulating tumor DNA (ctDNA).

Response 2: Thank you for your productive comment. We have added a sentence defining Circulating tumor DNA as follows:

(pp1, lines 26-27)

In addition, circulating tumor DNA (ctDNA) is a DNA fragment released into the blood by tumor cell-derived cell death or apoptosis.

Comments 3: A few words throughout the text are not separated by space. Please check.

Response 3: Thank you for pointing out. We have checked and corrected.

Round 2

Reviewer 3 Report

Comments and Suggestions for Authors

-